# Understanding self-assembly at molecular level enables controlled design of DNA G-wires of different properties

Daša Pavc [1,2], Nerea Sebastian [3], Lea Spindler [3,4], Irena Drevenšek-Olenik [3,5], Gorazd Koderman Podboršek [6,7], Janez Plavec [1,2,8] & Primož Šket [1✉]

A possible engineering of materials with diverse bio- and nano-applications relies on robust self-assembly of oligonucleotides. Bottom-up approach utilizing guanine-rich DNA oligonu-cleotides can lead to formation of G-wires, nanostructures consisting of continuous stacks of G-quartets. However, G-wire structure and self-assembly process remain poorly understood, although they are crucial for optimizing properties needed for specific applications. Herein, we use nuclear magnetic resonance to get insights at molecular level on how chosen short, guanine-rich oligonucleotides self-assemble into G-wires, whereas complementary methods are used for their characterization. Additionally, unravelling mechanistic details enable us to guide G-wire self-assembly in a controlled manner. MD simulations provide insight why loop residues with considerably different properties, i.e., hydrogen-bond affinity, stacking inter-actions, electronic effects and hydrophobicity extensively increase or decrease G-wire length. Our results provide fundamental understanding of G-wire self-assembly process useful for future design of nanomaterials with specific properties.

[1] Slovenian NMR Centre, National Institute of Chemistry, 1000 Ljubljana, Slovenia. [2] Faculty of Chemistry and Chemical Technology, University of Ljubljana, 1000 Ljubljana, Slovenia. [3] Department of Complex Matter, Jožef Stefan Institute, 1000 Ljubljana, Slovenia. [4] Faculty of Mechanical Engineering, University of Maribor, 2000 Maribor, Slovenia. [5] Faculty of Mathematics and Physics, University of Ljubljana, 1000 Ljubljana, Slovenia. [6] Department of Materials Chemistry, National Institute of Chemistry, 1000 Ljubljana, Slovenia. [7] Jožef Stefan International Postgraduate School, 1000 Ljubljana, Slovenia. [8] EN-FIST, Center of Excellence, 1000 Ljubljana, Slovenia. ✉email: primoz.sket@ki.si

dsDNA is an interesting molecule for nanoscience and nanotechnology, arguably due to its programmable self-assembling properties desired in bottom-up approaches. Simple Watson-Crick base pairing enables rational design of DNA nanostructures, ranging from static 2D to 3D materials, complex constructs as well as dynamic molecular machines with precise and controllable motions with various biotechnological and nanotechnological applications[1–4]. In comparison to dsDNA, four-stranded DNA exhibits interesting optical and electronic properties, superior resistance to enzymatic degradation as well as mechanical and thermal stability on substrates thus offering diverse applications[5–15]. An example of such four-stranded DNA structures are G-wires, which are long, G-quadruplex-based nanostructures composed of continuous runs of G-quartets. These planar motifs consist of four guanine residues held together by Hoogsteen-type of hydrogen bonds (Supplementary Fig. 1a). A unique feature of G-quartets is sensitivity to cations, which assist their formation and stacking into G-quadruplexes (Supplementary Fig. 1b) as well as potentially leading to G-quadruplex-based nanostructures. Furthermore, non-guanine residues often form loops, which are another important structural element of G-quadruplexes.

Since the first report in 1994 on G-wires formed by $d(G_4T_2G_4)$, they have been gaining interest due to their potential use as nanosensors, photonic and electronic nanodevices[6–10,16–20]. These long nanostructures can be formed by self-assembly of poly(G) strands as well as of short G-rich oligonucleotides[5,16,21–28]. Among proposed mechanisms for the latter are interlocking of G-quadruplexes via slipped G-rich strands (Supplementary Fig. 1c) and via sticky ends, e.g., GC-ends[16,22–25] (Supplementary Fig. 1d). Alternatively, G-wires can be formed via π–π stacking of G-quadruplexes[25–28] (Supplementary Fig. 1e). Fundamental understanding of self-assembly process and structure of G-wires remains poor, which is however needed to achieve precise controllable as well as tunable formations and therefore tailoring of materials with the desired properties. The first step towards better understanding was achieved only recently, where solution-state atomic force microscopy (AFM) complemented with molecular dynamics (MD) simulations were used to show that $d(G_4T_2G_4)$ can self-assemble into three different structural types of G-wires via interlocking of slipped G-rich strand mechanism[22]. However, study of G-wire self-assembly still remains challenging since spontaneous multimerization makes difficult to "catch" individual building blocks (i.e., G-quadruplexes) in order to characterize them and monitor their further assembly in a controlled manner.

Herein, we offer detailed insights into the self-assembly process of $d(G_2AG_4AG_2)$ G-wire formation at molecular level. Previously, we showed that GC ends attached to either 5′-end or both 5′ and 3′ ends of oligonucleotide $d(G_2AG_4AG_2)$ hindered multimerization[29]. In general, such type of oligonucleotides, i.e., $G_2X(G_4X)_nG_2$ (where X = T, A, TC and $n = 0, 1, 2,$ and 4) with 5′-GC and/or 3′-GC ends, were subject of many studies for their multimerization ability[23–25,30,31]. Varying factors such as sample preparation and cation concentration led us to solution conditions, where single G-wire building blocks (G-quadruplexes) are dominant, thus enable their structural determination by nuclear magnetic resonance (NMR). We believe that beside steric availability of terminal G-quartets, hydrophobic properties of stacking interfaces are important in aqueous solutions and thus have a key role in self-assembly of stacked G-wires. Therefore, G-wires lengths can be manipulated by designing oligonucleotides with different loop residues near terminal G-quartets. Significant changes might be expected already by substituting adenine for thymine and cytosine residues, whereas considerably different, i.e., in terms of hydrogen-bond affinity, stacking properties, electronic effects and hydrophobicity, c3 linkers and abasic residues could influence lengths of G-wires even more. For

visualization and characterization of G-wires we used several complementary methods including dynamic light scattering (DLS), AFM, scanning and transmission electron microscopies (SEM, TEM).

## Results and discussion

**$d(G_2AG_4AG_2)$ forms G-wires.** We probed $d(G_2AG_4AG_2)$ oligonucleotide for G-wire formation by slow cooling (annealing) its 1.0 mM solution in 100 mM KCl (see "Method 1" in "Methods" section). A broad hump between $\delta$ 9.7 and 11.4 ppm in 1D $^1H$ NMR spectrum at 25 °C was tentatively assigned to guanine H1 protons of G-wires (Fig. 1a). In agreement, DLS revealed translation diffusion coefficient $D_t$ of $0.33 \times 10^{-10}$ m²s⁻¹, which corresponds to G-wires with 43 nm effective length (Fig. 1b). AFM images at 0.2 μM $d(G_2AG_4AG_2)$ concentration (see "Methods" section, Fig. 1c and Supplementary Fig. 2a) show rod-like structures with 2.0 nm average height as typical for G-wires[23,26,32]. Length analysis shows a log-normal distribution with 22.1 nm mean value and 65 nm maximum detected length. At higher, 2 μM $d(G_2AG_4AG_2)$ concentration AFM shows an interconnected network of G-wires with 2.4 nm height (Supplementary Fig. 2b, c). Furthermore, SEM and TEM images at 1.0 mM $d(G_2AG_4AG_2)$ concentration show randomly distributed G-wires and their bigger deposits (Fig. 1d–f and Supplementary Figs. 3 and 4). Comparative to SEM, TEM offers more discernible images of shorter G-wires (Fig. 1e, f and Supplementary Fig. 4). The width of G-wires is between 10 and 20 nm (Fig. 1g), which is greater than solution diameter of G-quadruplexes (~2.8 nm), but in agreement with previously observed width by AFM[28]. Various observed lengths of $d(G_2AG_4AG_2)$ G-wires by SEM/TEM, AFM, and DLS can be explained by different surface and solution behavior as well as sample concentration. These data demonstrate that $d(G_2AG_4AG_2)$ is suitable model for in depth mechanistic study of G-wire self-assembly.

**Structural rearrangement is the crucial step in self-assembly of $d(G_2AG_4AG_2)$ G-wires.** By lowering temperature to 0 °C and varying KCl concentration (100–3 mM) we determined solution conditions where Q1k and Q2k, first and second G-quadruplexes in $d(G_2AG_4AG_2)$ G-wire self-assembly are predominant structures (see "Method 2" in "Methods" section; Fig. 2 and Supplementary Figs. 5 and 6). We used fast cooling (quenching) of $d(G_2AG_4AG_2)$ in 3 mM KCl to shift equilibrium towards Q1k (see "Method 3" in "Methods" section). 1D $^1H$ NMR spectrum of Q1k exhibits eight sharp signals between $\delta$ 10.77 and 11.67 ppm, which correspond to H1 protons of guanines in G-quartets (Fig. 2a). Eight H1 proton signals together with $D_t$ of $1.55 \times 10^{-10}$ m² s⁻¹ and mobility on a native polyacrylamide electrophoresis (PAGE) gel indicate that Q1k is a dimeric G-quadruplex composed of four G-quartets (Fig. 2a and Supplementary Figs. 7 and 8). Unambiguous assignment of H1 and H8 guanine proton resonances revealed two unusual chemical shifts $\delta_{G2H1}$ at 10.77 and $\delta_{G1H8}$ at 6.28 ppm, where the latter indicates presence of A(GGGG)A hexad[29,33] (Supplementary Fig. 9). Hereupon, $\delta_{G2H1}$ and $\delta_{G1H8}$ have been exploited as NMR spectroscopic markers, which provide an easy way to distinguish between Qk and Qt structural types (Qk- and Qt-type, vide infra) in the course of $d(G_2AG_4AG_2)$ multimerization. Examination of 2D NOESY spectrum reveals that Q1k is C2 symmetric G-quadruplex formally composed of two units: 5′-antiparallel and 3′-parallel (Fig. 2a and Supplementary Fig. 10). The former is made of G1-G5-G1-G5 and G2-G4-G2-G4 quartets and two single-nucleotide (A3) lateral loops, whereas the latter consists of G7-G10-G7-G10 quartet and A8-(G6-G9-G6-G9)-A8 hexad, where A8 residues constitute propeller loops (Fig. 2a). The corresponding circular dichroism (CD) spectrum with minimum

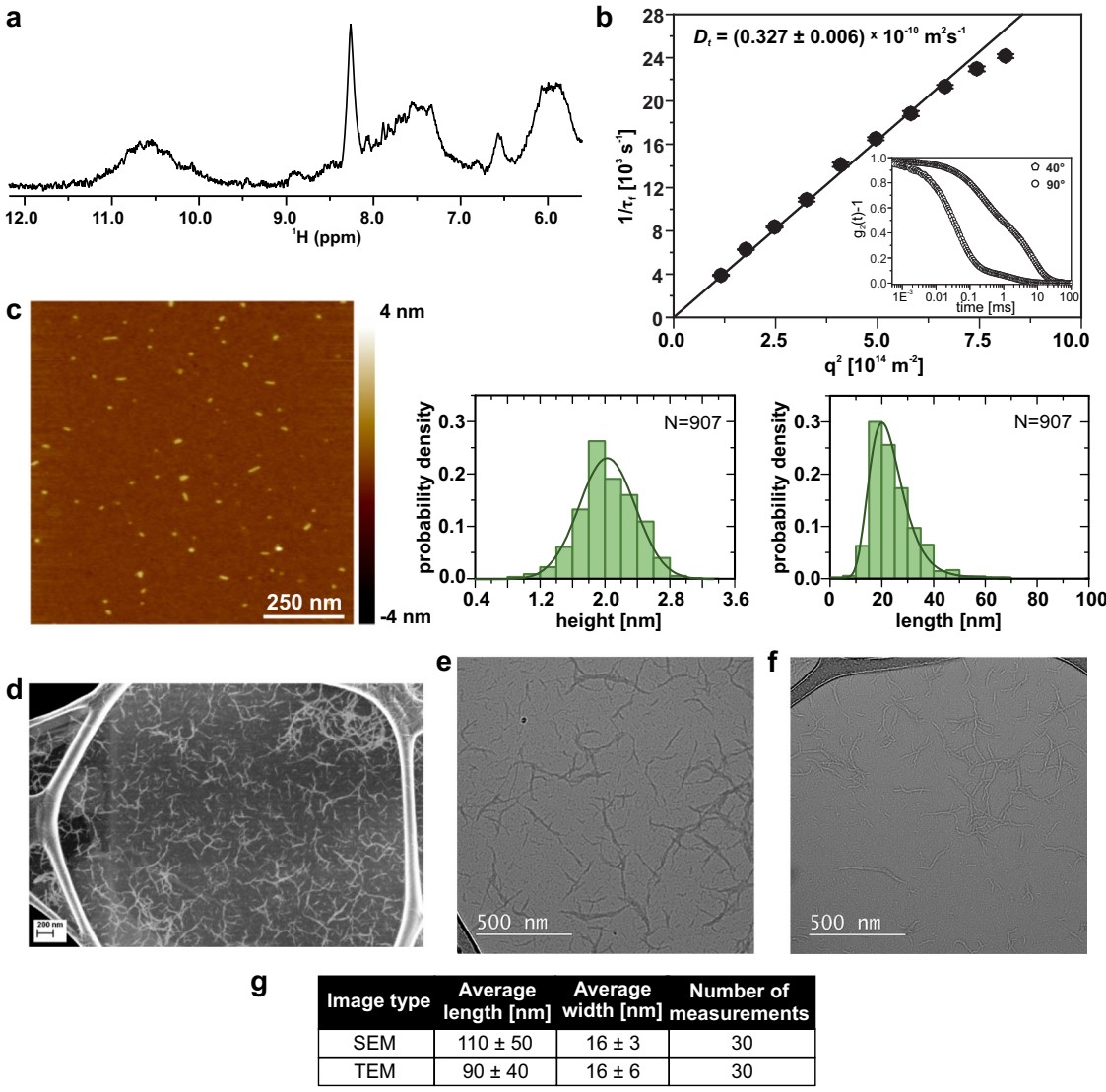

**Fig. 1 G-wires formed by d(G₂AG₄AG₂). a** Selected region of 1D $^{1}$H NMR spectrum shows broad signals, typical for G-wires. **b** DLS data. Insert shows experimental autocorrelation curves. **c** Representative AFM height image of G-wires with corresponding length and height histograms. AFM image was obtained at very low oligonucleotide concentration (0.2 μM) enabling detection of individual wires, with mean height of 2.0 ± 0.1 nm and mean length of 22.1 ± 0.4 nm. Statistics collected over eight images. **d** SEM images and conventional TEM images of unstained G-wires taken at substantial **e** under-focus and **f** over-focus. We observe random distribution of G-wires on GO support. Smaller G-wires are more clearly visible in TEM images. **g** Table of estimated average length and width of G-wires from SEM and TEM images. Sample was assembled via method 1 (see "Methods" section).

at 240 nm, maximum at 270 nm and shoulder at 300 nm is in agreement with the mixed antiparallel/parallel folding topology of Q1k (Fig. 2a). Q1k is transformed into Q2k as followed at 0 °C, where the rate of Q2k formation varies with KCl concentrations (Fig. 2b and Supplementary Figs. 5 and 6). Quenching of d(G₂AG₄AG₂) in 60 mM KCl shifts equilibrium towards Q2k and thus enables its NMR structural analysis (see "Method 4" in "Methods" section). Compared to Q1k, 1D $^{1}$H NMR spectrum of Q2k exhibits upfield shifted H1 signals as well as lower $D_t$ of 1.28 × $10^{-10}$ m$^2$ s$^{-1}$ and slower mobility on native PAGE gel, which suggest formation of bigger structure for the latter G-quadruplex (Fig. 2c and Supplementary Figs. 8 and 11). NMR spectroscopic markers $\delta_{G2H1}$ at 10.45 and $\delta_{G1H8}$ at 5.96 ppm together with cross-peaks in 2D NOESY spectrum show that Q2k exhibits the same combination of structural elements as Q1k (Fig. 2c and Supplementary Figs. 12 and 13). Changes in chemical shifts of H1 protons are the most pronounced for G7 and G10 ($\Delta\delta_{G7H1}$ of 0.44 and $\Delta\delta_{G10H1}$ of 0.91 ppm), indicating 3′-3′ stacking of individual

G-quadruplex building blocks (Fig. 2c and Supplementary Figs. 9 and 12). 3′-3′ stacking in Q2k is further supported by H-D exchange experiment, where H1 protons of the outer G2-G4-G2-G4 quartet disappear immediately after dissolving in D₂O. The apparent rate of exchange is slower for remaining G-quartets, where their H1 signals are still observable in 1D $^{1}$H NMR spectrum after 11 days (Fig. 2c). Comparison of Q1k and Q2k CD spectra reveals highly similar shapes with increase in shoulder around $\lambda$ 300 nm for the latter, which can be attributed to 3′-3′ stacking[34] (Fig. 2a, c).

We observed a new set of signals in 1D $^{1}$H NMR spectra of d(G₂AG₄AG₂) 6 days after addition of 3 mM KCl at 0 °C corresponding to Q2t (Supplementary Fig. 14a, c). Equilibrium is shifted towards formation of Q2t when 3 mM KCl is added into solution of d(G₂AG₄AG₂) at 25 °C (Fig. 2d and Supplementary Fig. 14b, d). Q2t becomes dominant structure at 25 °C, when d(G₂AG₄AG₂) is annealed in 15 mM KCl (see "Method 5" in "Methods" section; Fig. 2e). In 1D $^{1}$H NMR spectrum of Q2t H1

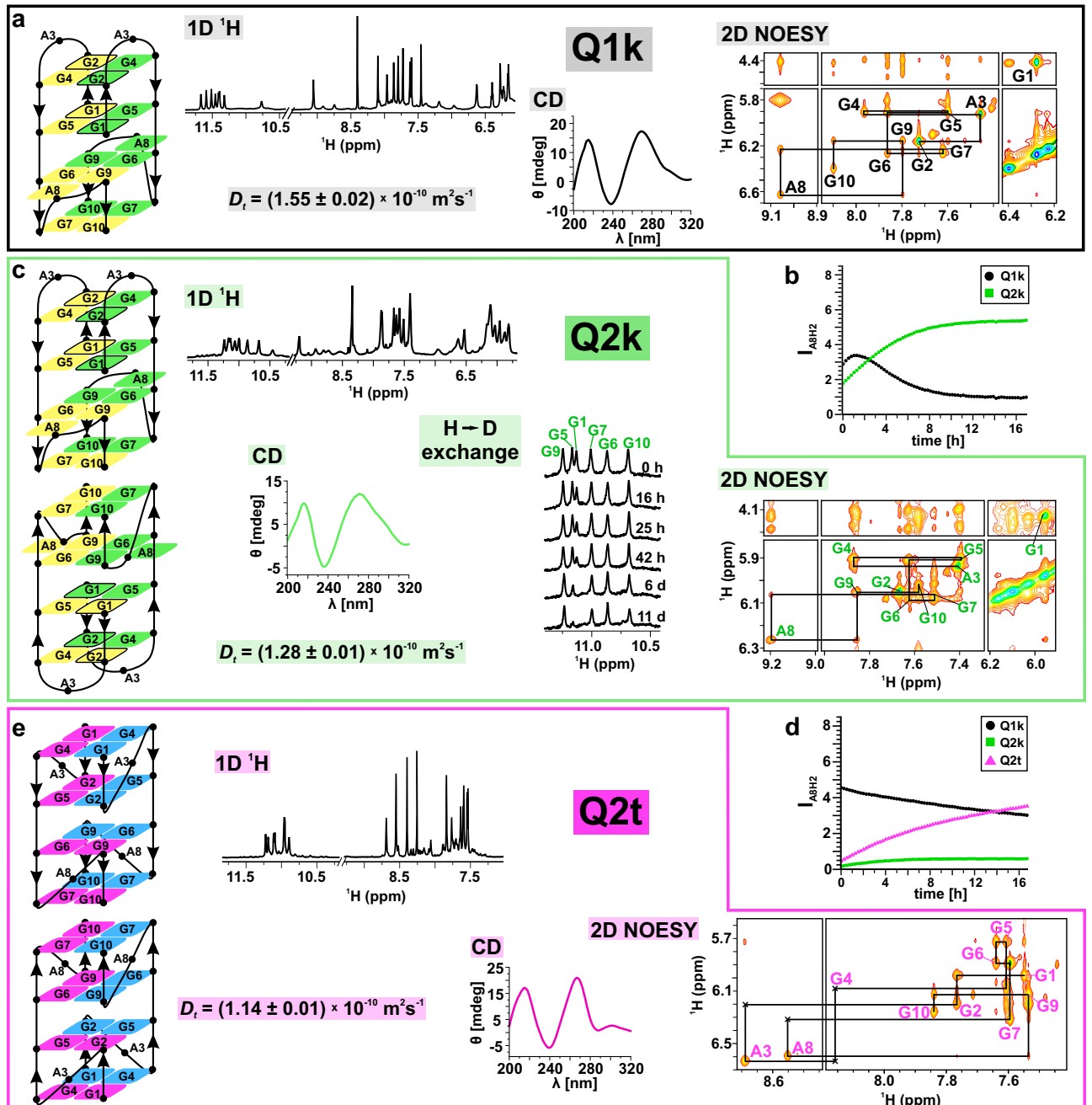

**Fig. 2 Initial steps of d(G₂AG₄AG₂) G-wire self-assembly. a** NMR data, CD spectrum and folding topology of Q1k. In 2D NOESY spectrum ($\tau_m$ 150 ms) the intra-nucleotide H8$_{(n)}$-H1′$_{(n)}$ cross-peaks are labeled with corresponding residue numbers, where their observed intensities reveal *syn* conformation along glycosidic bond for G1 and G2 and *anti* for the remaining residues. Black lines show sequential walk, which is followed from G2 up to G10 residue via H8$_{(i)}$-H1′$_{(i)}$ and H1′$_{(i)}$-H8$_{(i+1)}$ connectivities as characteristic for *anti-anti* step. G1 and G2 residues in *syn* conformation are presented with rectangles surrounded with black in Q1k folding topology. Arrows indicate progression of strand from 5′→3′ end. **b** Q1k→Q2k transformation at 0 °C in 100 mM KCl. **c** Folding topology of Q2k and corresponding NMR and CD data. Similarly to Q1k, sequential walk in Q2k is followed from G2 up to G10 residues in 2D NOESY spectrum ($\tau_m$ 150 ms). Deuterium exchange experiment supports 3′-3′ stacking in Q2k. **d** Q1k→Q2k→Q2t transformation at 25 °C in 3 mM KCl. **e** Structural differences between Q2t and Q1k/Q2k are reflected in CD, 1D ¹H and 2D NOESY ($\tau_m$ 150 ms) spectra. Crosses in 2D NOESY denote weak cross-peaks. Simplified scheme of these structures is presented in Fig. 3. Samples were assembled via **a** method 3, **b** method 2, **c** method 4, **d** method 2, and **e** method 5 (see "Methods" section).

and H8 of guanines, including G2H1 and G1H8 resonate within narrower chemical shift range in comparison to Qk-type, indicating considerably different structure (Fig. 2e and Supplementary Fig. 15). $D_t$ of $1.14 \times 10^{-10}$ m² s⁻¹ and mobility on native PAGE gel suggest that Q2t is also composed of two stacked G-quadruplexes (Fig. 2e and Supplementary Figs. 8 and 16). Perusal of 2D NOESY spectra reveals formation of G1-G4-G1-G4

and G2-G5-G2-G5 quartets in Q2t, which together with G6-G9-G6-G9 and G7-G10-G7-G10 quartets establish all-parallel folding topology consisting of eight G-quartets with A3 and A8 residues in propeller loops (hereupon referred as loop residues at position 3 and 8) (Fig. 2e and Supplementary Fig. 17). 3′-3′ stacking in Q2t is confirmed by inter-quadruplex G7H1-G7H8 and G10H1-G10H8 NOEs (Supplementary Fig. 17b). Q2t displays CD

spectrum typical of all-parallel G-quadruplexes and a peak at $\lambda$ 300 nm reflecting 3'-3' stacking (Fig. 2e).

Since Q2k and Q2t are both composed of two stacked G-quadruplexes the question is whether there are two distinct pathways, where Q1k can be transformed to both Q2k and Q2t, or alternatively there is a straight pathway, where Q1k, Q2k and Q2t are formed subsequently. Therefore, we prepared two d(G$_2$AG$_4$AG$_2$) samples in 15 mM KCl and tested the effect of quenching and annealing. 1D $^1$H NMR spectra obtained at 0 °C on the same day as samples were prepared, reveal presence of Q1k and Q2k in the quenched sample, while only Q2t is observed in the annealed one (Supplementary Fig. 18). Therefore, Q1k and Q2k are kinetically preferred (Qk-type), whereas formation of Q2t (Qt-type) is thermodynamically favored. In agreement, 1D $^1$H NMR spectra of both samples acquired four weeks later, exhibit only signals corresponding to Q2t (Supplementary Fig. 18).

**d(G$_2$AG$_4$AG$_2$) G-wires are structurally homogenous**. Q2t exhibits terminal 5'-quartets, which offer possibility for further stacking. 5'-5' stacking of two Q2t leads to formation of Q4t, which is detected already at 15 mM KCl and 25 °C (Supplementary Figs. 15 and 19a). Qt-type for Q4t is supported by its similar $\delta_{G1H8}$ as observed for Q2t (Supplementary Fig. 15). Two sets of signals for H1 and H8 protons of all guanines in NMR spectra of Q4t are attributed to two distinct shielding environments of atoms, e.g., G1-G4-G1-G4 quartets, where two of them are terminal 5'-quartets, whereas the remaining two form 5'-5' stacking interface (Supplementary Figs. 15 and 19a). Terminal 5'-quartets of Q4t are crucial for subsequent multiple 5'-5' stacking leading to formation of d(G$_2$AG$_4$AG$_2$) G-wires (Fig. 3).

To confirm that G-wires are Qt-type, i.e., are structurally homogeneous and are formed via described pathway, we prepared d(G$_2$AG$_4$AG$_2$) with isotopically enriched G1 residue in 40 and 80 mM KCl with different heat treatments. 2D $^{13}$C-edited HSQC spectra acquired after preparation of samples without heat treatment and with quenching exhibit $\delta_{G1H8}$ of Q2k, whereas one week later only $\delta_{G1H8}$ of Qt-type is observed (Supplementary Fig. 19b–d). Contrary, for d(G$_2$AG$_4$AG$_2$) annealed in 80 mM KCl only cross-peaks of Qt-type are observed in HSQC spectrum already from the beginning (Supplementary Fig. 19e).

These results confirm that irrespective of KCl concentration d(G$_2$AG$_4$AG$_2$) folds firstly into Qk-type G-quadruplexes, which is followed by rearrangement and formation of Qt-type of structures. Rearrangement from Q2k into Q2t occurs via long-lived intermediate, which will be described in detail in a separate study. Interestingly, such spontaneous rearrangement as observed for d(G$_2$AG$_4$AG$_2$) self-assembly enables formation of stacked G-wires, which have been hard to obtain since multimerization of G-quadruplexes usually stops at the 5'-5' dimer stage despite free 3'-quartets[35]. Lately, an approach to avoid this problem was developed, i.e., incorporation of 3'-3' inversion of polarity sites, where G-wire self-assembly is guided solely via 5'-5' stacking[27,28]. We have showed that stacked G-wires can be obtained also using standard nucleotides on d(G$_2$AG$_4$AG$_2$) G-wire, where self-assembly process consists of following steps: I) folding of Q1k, II) 3'-3' stacking of Q1k (formation of Q2k), III) structural rearrangement of Q2k (formation of Q2t), IV) 5'-5' stacking of Q2t (formation of Q4t) and consequent V) multiple 5'-5' stacking of Q2t/Q4t leading to final Qnt G-wires composed of $n$ number of stacked Qt-type G-quadruplexes (Fig. 3).

There are two rate-determining steps of d(G$_2$AG$_4$AG$_2$) G-wire self-assembly, which depend on the temperature. At 0 °C, formation of Q2t determines the overall rate of self-assembly, since the system

does not have enough energy for an immediate structural transition of Q2k to Q2t, whereas at 25 °C and above, structural transition occurs rather quickly, therefore the formation of Q2k, which depends on cation concentration is the rate-determining step of d(G$_2$AG$_4$AG$_2$) G-wire self-assembly (Supplementary Figs. 5 and 6 and 14c, d). Since there are different structures involved in self-assembly pathway we prepared a simplified phase diagram showing how concentration of cations and oligonucleotide shifts equilibrium towards formation of G-wire (phase diagram sample preparation, Methods; Supplementary Fig. 20). As expected, higher concentrations of oligonucleotides or cations facilitate formation of G-wires, i.e., at 0.1 mM d(G$_2$AG$_4$AG$_2$) concentration G-wires are observed at 80 mM KCl, while already 15 mM KCl is able to induce G-wire self-assembly at 2 mM oligonucleotide concentration (Supplementary Fig. 20).

**Design of G-wires of different lengths**. We expect that loop residues might play an important role in determining the properties of G-wires, e.g., length. Therefore, we systematically substituted adenines with thymines and cytosines at positions 3 and/or 8 (Fig. 4a). All modified oligonucleotides are capable of G-wire formation, as indicated by a broad hump observed in H1 region of their 1D $^1$H NMR spectra as well as ladder pattern with smearing on native PAGE gel (see "Method 1" in "Methods" section; Supplementary Figs. 21 and 22a). Modified oligonucleotides follow the same folding pathway as parent d(G$_2$AG$_4$AG$_2$) (Fig. 3) as we demonstrated by adding KCl to final 3 mM concentration into solution of d(G$_2$TG$_4$AG$_2$), d(G$_2$CG$_4$AG$_2$), d(G$_2$AG$_4$TG$_2$), d(G$_2$AG$_4$CG$_2$), d(G$_2$TG$_4$TG$_2$) and d(G$_2$CG$_4$CG$_2$) oligonucleotides and following changes by NMR at 25 °C. 1D $^1$H NMR spectra recorded immediately after addition of KCl exhibit multiple sets of signals between $\delta$ 10.5 and 12.3 ppm, likely corresponding to H1 protons of guanines in G-quartets of Q1k, Q2k and Q2t, whereas only one set of signals is dominant in 1D $^1$H NMR spectra one day later (Supplementary Fig. 23). Furthermore, G-wires formed by modified oligonucleotides exhibit highly similar CD spectra as parent d(G$_2$AG$_4$AG$_2$) G-wire, which supports formation of Qt-type building blocks in all cases (Supplementary Fig. 24). DLS revealed $D_t$ between 0.25 and 0.44 × 10$^{-10}$ m$^2$ s$^{-1}$ for modified oligonucleotides indicating formation of rather long G-wires (Fig. 4a and Supplementary Fig. 22b). Using theory for cylindrical scatterers[36], the $D_t$ correspond to G-wires with effective lengths between 28 and 67 nm (Fig. 4a and Supplementary Fig. 22c). Based on our determined G-quadruplex model and with a typical stacking distance between G-quartets of 0.34 nm, the number of stacked Qt-type building blocks ($n$) in Qnt-type G-wires ranges from 20 to 49 (Figs. 3 and 4a).

The difference in length of d(G$_2$AG$_4$AG$_2$) and the longest d(G$_2$AG$_4$CG$_2$) G-wires is clearly observed in AFM images, where the latter exhibit 31.6 nm mean length with values up to 120 nm (Fig. 4b and Supplementary Fig. 25a). For d(G$_2$AG$_4$CG$_2$) we were also able to obtain AFM images of G-wires deposited on mica, which was not pre-treated with MgCl$_2$ where structures have higher mean length of 40.6 nm and values up to 160 nm indicating that substrate pre-treatment allows adhesion of the smaller G-wires, which are not observed otherwise (Supplementary Fig. 25b, c). In agreement, SEM and TEM images of d(G$_2$AG$_4$CG$_2$) show G-wires with average length values up to 180 nm, which are clearly longer compared to parent ones (Fig. 4c–f and Supplementary Figs. 26 and 27).

**The type of loop residues influences thermal stability of G-wires**. We evaluated thermal stability of G-wires by following their unfolding from 5 to 85 °C by NMR melting experiment

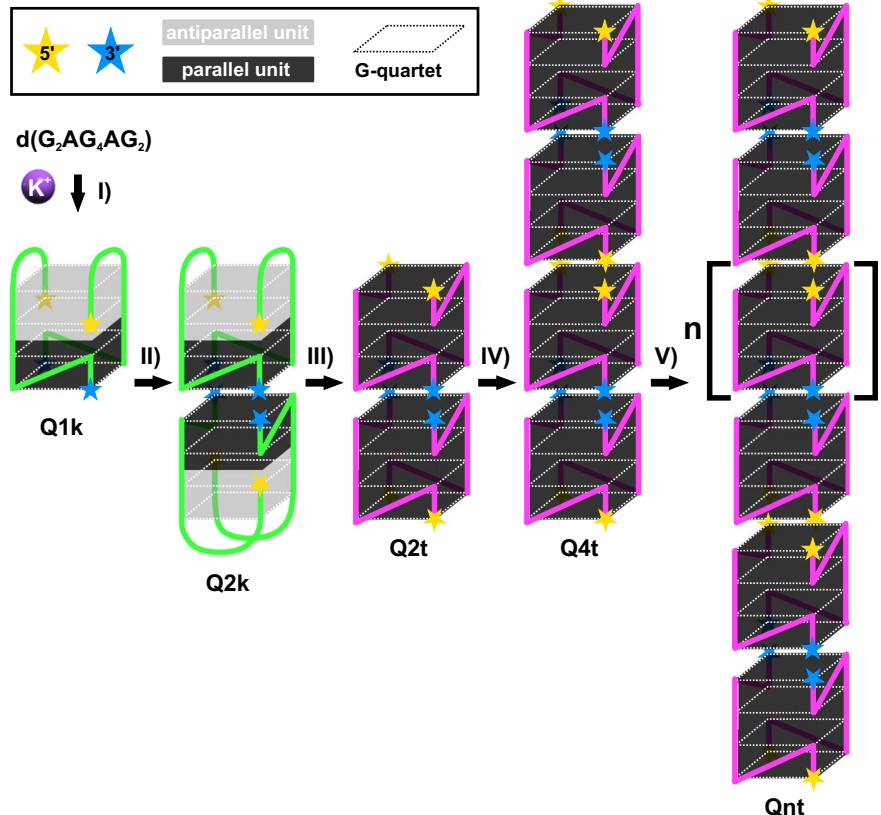

**Fig. 3 A cartoon representation of d(G₂AG₄AG₂) G-wire self-assembly.** The process of d(G₂AG₄AG₂) G-wire formation starts with **I** folding of Q1k and is followed by **II** 3′-3′ stacking, which results in formation of Q2k. Both *Qk*-type G-quadruplexes consist of 5′-antiparallel unit (gray) and 3′-parallel unit (black). **III** Structural rearrangement of Q2k results in formation of an all-parallel Q2t. Formation of Q2t is followed by **IV** 5′-5′ stacking, which leads to formation of Q4t. Subsequent **V** multiple 5′-5′ stacking result in formation of Qnt G-wires with different length and *n* number of stacked Qt-type G-quadruplexes. Yellow and blue stars mark the 5′-end and 3′-end of G-quadruplexes, respectively. Dotted rectangles present G-quartets.

(Supplementary Figs. 28 and 29). 1D $^1$H NMR spectra of G-wires acquired between 5 and 25 °C exhibit a broad hump in H1 region, while broad signals on top of a hump start to appear at 35 °C, which indicates beginning of G-wires dissociation into shorter structures. Above 55 °C a set of sharp signals is observed in addition to broader signals, which likely correspond to Q2t. At 85 °C only sharp signals are observed in 1D $^1$H NMR spectra of all G-wires indicating that solely Q2t is stable at this temperature, as later confirmed by NMR melting experiment on a sample with 25 mM KCl, where only Q2t and Q4t are present in solution at room temperature (Supplementary Fig. 30).

It should be mentioned that the thermal stability of these structures greatly depends on the concentration of KCl, i.e., at 85 °C, Q2t is still observed in 100 mM KCl, but is completely unfolded in the presence of 25 mM KCl. When cooling the sample with 25 mM KCl back from 85 to 65 °C, H1 region of 1D $^1$H NMR spectrum exhibits signals that correspond to Q2t and in a lesser amount to Q1k indicating that at higher temperature the equilibrium is shifted towards formation of Q2t (Supplementary Fig. 30). Interestingly, Q2k is not observed at 65 °C, likely due to immediate structural rearrangement of Q2k to Q2t, which is faster at higher temperatures as shown before at 0 and 25 °C (Supplementary Figs. 14c, d and 30). Returning back at 25 °C we observe H1 protons signals corresponding to Q2t and Q4t as well as to Q1k and Q2k, which indicates the same folding and refolding pathway of d(G₂AG₄AG₂). Reversibility of self-assembly process was also observed for all studied G-wires.

UV melting curves obtained from 5 up to 95 °C on G-wires exhibit one transition, which actually corresponds to unfolding of

Q2t, which is the last stable G-quadruplex in unfolding of G-wires as demonstrated before by NMR melting experiments (Supplementary Figs. 28–31).

Since UV melting curves did not reach a clear plateau at 95 °C, mid-transition temperatures ($T_{1/2}$) are only estimations, where most of them are in the range between 83 and 92 °C (Supplementary Fig. 31d). The most stable are d(G₂AG₄TG₂) G-wires where $T_{1/2}$ exceeds 92 °C (Supplementary Fig. 31d). Plot, where G-wires are arranged by increasing $T_{1/2}$ reveals that their thermal stability greatly depends on the type of loop residue at position 8 (Supplementary Fig. 31e). G-wires with T and A at position 8 are the most and the least thermally stable, respectively, which correlates with previously observed data on single G-quadruplexes[37–39].

**Why can the type of loop residues affect the length of G-wires?** In aqueous solution nonpolar molecules tend to self-assemble due to the hydrophobic effect. It was shown that 5′-quartets are more hydrophobic compared to 3′-quartets, which might add to 5′-5′ stacking being the preferred mode of G-quadruplex multimerization[40]. For substitutions with the same type of loop residues at positions 3 and 8 lengths, of G-wires increase as follows: d(G₂TG₄TG₂) < d(G₂AG₄AG₂) < d(G₂CG₄CG₂) (Supplementary Fig. 22c). In accordance, the hydrophobicity of nucleobases decreases in the same order, i.e., T > A > C[41]. Therefore, our observations suggest that hydrophobicity of loop residues affects nearby G-quartets and in this way influence length of G-wires. In complete agreement, folding of oligonucleotides with c3 linker (li) and abasic residue (ab), which are

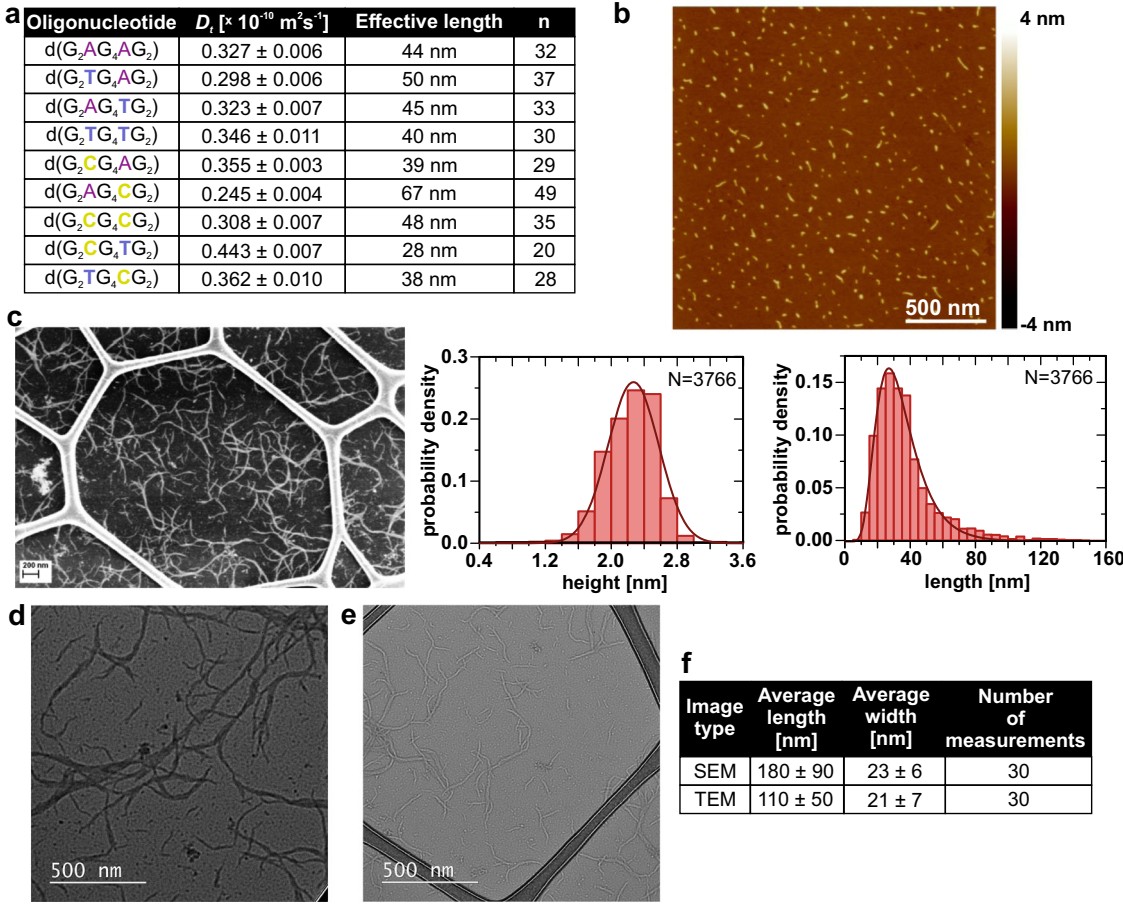

**Fig. 4 A single loop residue substitution considerably extends G-wires length. a** Table shows $D_t$, effective lengths and number of stacked Qt-type building blocks ($n$) for Qnt-type G-wires obtained from DLS. **b** Representative AFM height image of G-wires formed by d($G_2AG_4\underline{C}G_2$) with corresponding length and height histograms. Low deposition concentration allows visualization of individual d($G_2AG_4\underline{C}G_2$) G-wires with 2.3 ± 0.1 nm mean height and mean length 31.6 ± 0.3 nm. Statistics collected over 12 images. **c** SEM image of G-wires shows their random distribution on GO support. We see that modified d($G_2AG_4\underline{C}G_2$) G-wires are clearly longer than parent ones. **d**, **e** Conventional TEM images of unstained d($G_2AG_4\underline{C}G_2$) G-wires. Images were taken at substantial **d** under-focus and **e** over-focus. Smaller G-wires are more clearly visible in **e**. **f** Average length and width of d($G_2AG_4\underline{C}G_2$) G-wires estimated from SEM and TEM images. Samples were assembled via method 1 (see "Methods" section).

more and less hydrophobic, respectively compared to T, A, and C leads to formation of the shortest (15 nm) and the longest (>90 nm) G-wires as confirmed by NMR and DLS (Fig. 5a–c and Supplementary Fig. 32). However, one must be aware that nucleotides also possess different rotational degrees of freedom, i.e., the c3 linker exhibits significantly greater rotational freedom compared to deoxyribose in the backbone, which can impact the equilibrium distribution of the different guanine-rich DNA folds and consequently affect length of resulting G-wires.

To gain further insights into how loop residues affect G-wire lengths we performed NMR-derived structural calculations of d($G_2\underline{li}G_4\underline{li}G_2$), d($G_2TG_4TG_2$), d($G_2AG_4AG_2$), d($G_2CG_4CG_2$) and d($G_2\underline{ab}G_4\underline{ab}G_2$) Q4t G-quadruplexes (Fig. 6a–e and Supplementary Fig. 33). Analysis of intra-quadruplex and inter-quadruplex groove dimensions reveals that d($G_2\underline{ab}G_4\underline{ab}G_2$) Q4t exhibits the smallest size deviation with the nicest periodicity (Fig. 6e). Contrary, bigger deviations of intra- and inter-quadruplex groove dimensions are observed for d($G_2\underline{li}G_4\underline{li}G_2$), d($G_2TG_4TG_2$), d($G_2AG_4AG_2$), and d($G_2CG_4CG_2$) Q4t. It seems that more hydrophobic loop residues tend to interact with each other within the structure, which affects the dimensions of the grooves and consequently reduces the flexibility and distorts the planarity of G-quartets and thus makes formation of G-wires more difficult. In addition, structural calculations show that T, A, and C residues

can stack on the terminal 5′-quartets and thus interfere with further stacking (Fig. 6b–d and Supplementary Fig. 33). Incorporation of $\underline{ab}$ residues at only one position, i.e., 3 or 8 lead to formation of generally long G-wires as well (Fig. 5b, c). Interestingly, oligonucleotides with $\underline{li}$ residues only at one position yield quite long G-wires as well, which can be attributed to reduced hydrophobic interactions of loop residues and stacking on terminal 5′-quartets compared to bulkier residues.

Herein, using NMR we revealed the exact mechanism of how short, guanine-rich oligonucleotide self-assemble into G-wire at molecular level. The crucial step of this sophisticated mechanism includes structural rearrangement of kinetically favored G-quadruplex building block into thermodynamic one. Furthermore, we showed that properties of resulting G-wires, i.e., length and thermal stability can be tailored by changing the type and consequently features of loop residues. Using MD simulations we provided insights into the rationale behind ability of loop residues to influence G-wire properties, which include beside different loop residue hydrophobicity, also their interactions between loops as well as with outer G-quartets, crucial for structural integrity of G-quadruplexes and consequently final G-wires. We believe that findings of the present research will have a great impact on future design of oligonucleotides for G-wire formation and their functionalization needed for specific applications. Our discoveries

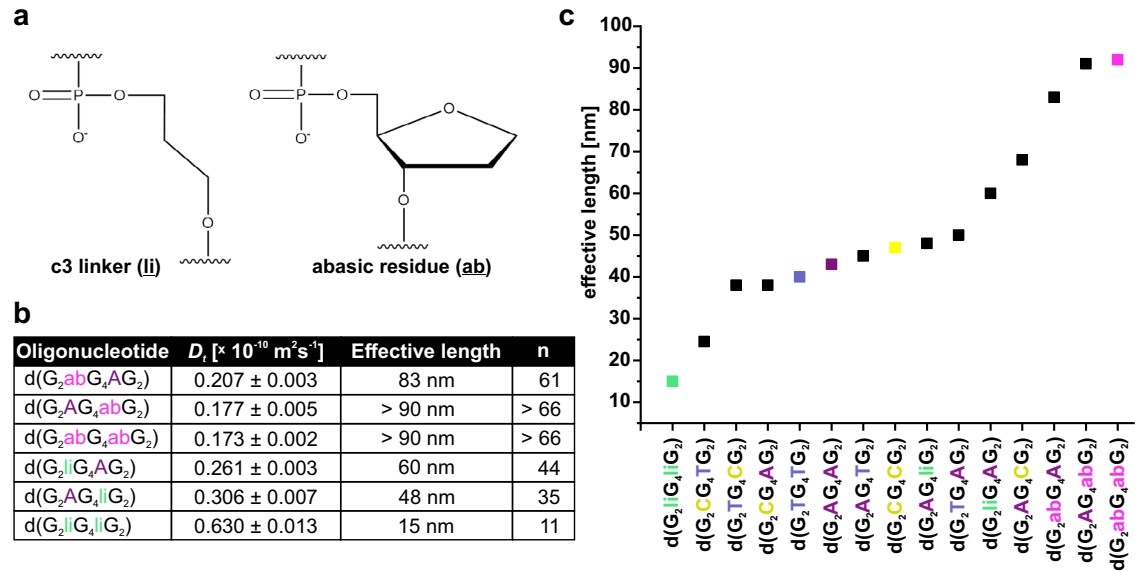

**Fig. 5 C3 linker and abasic nucleotide in loops influence the length of G-wires even more. a** Chemical structures of c3 linker (li) and abasic residue (ab). **b** A table showing $D_t$ obtained from DLS measurements and calculated effective lengths and number of stacked Qt-type building blocks. **c** In plot, oligonucleotides are arranged by increasing effective length of resulting G-wires (from left to right). For visualization in plot we used 92 and 91 nm as effective lengths of the longest two G-wires d(G₂abG₄abG₂) and d(G₂AG₄abG₂), whose lengths exceed the validity region of the used model. Samples were assembled via method 1 (see "Methods" section).

| Oligonucleotide | $D_t$ [× 10⁻¹⁰ m²s⁻¹] | Effective length | n |
|---|---|---|---|
| d(G₂abG₄AG₂) | 0.207 ± 0.003 | 83 nm | 61 |
| d(G₂AG₄abG₂) | 0.177 ± 0.005 | > 90 nm | > 66 |
| d(G₂abG₄abG₂) | 0.173 ± 0.002 | > 90 nm | > 66 |
| d(G₂liG₄AG₂) | 0.261 ± 0.003 | 60 nm | 44 |
| d(G₂AG₄liG₂) | 0.306 ± 0.007 | 48 nm | 35 |
| d(G₂liG₄liG₂) | 0.630 ± 0.013 | 15 nm | 11 |

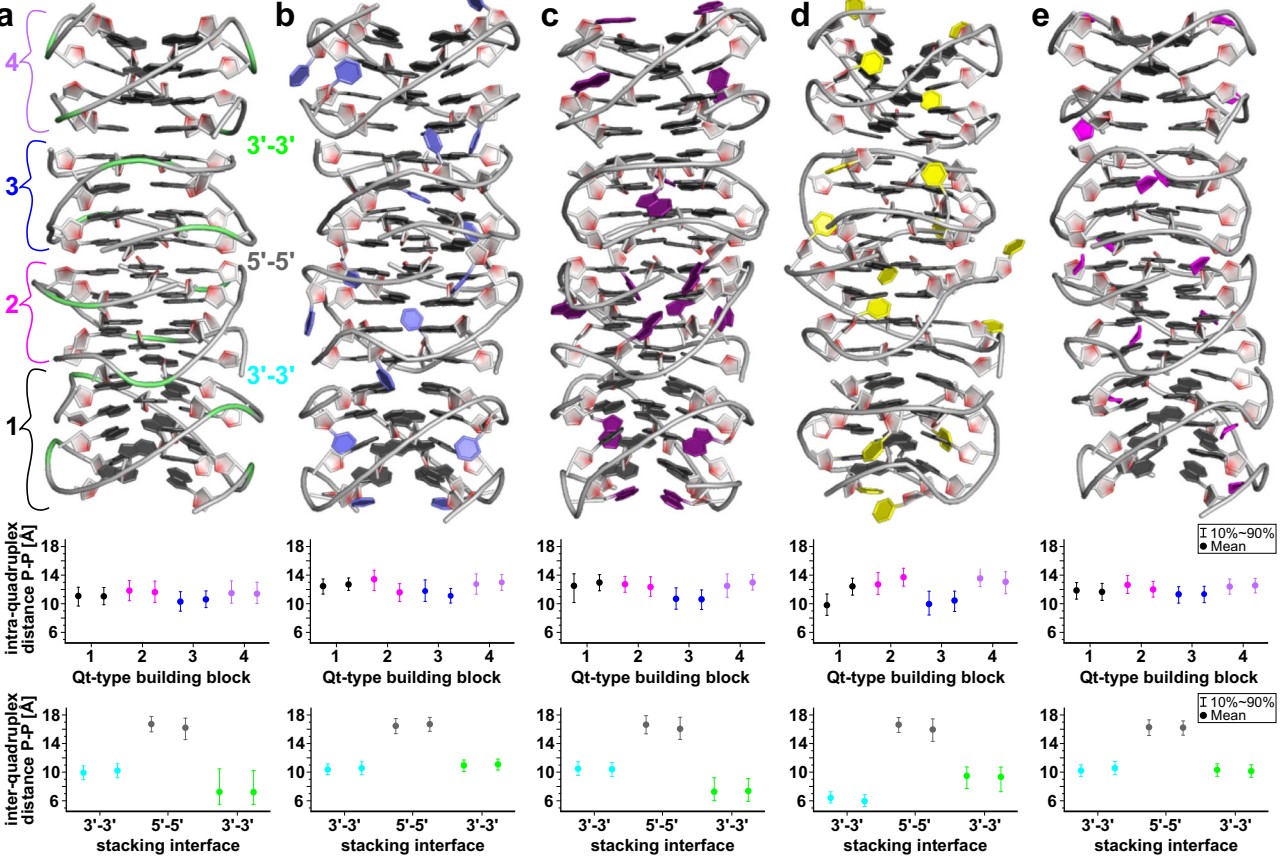

**Fig. 6 Hydrophobicity of loop residues and their stacking ability might explain differences in lengths of G-wires.** Q4t structures formed by **a** d(G₂liG₄liG₂), **b** d(G₂TG₄TG₂), **c** d(G₂AG₄AG₂), **d** d(G₂CG₄CG₂), and **e** d(G₂abG₄abG₂). Plots in **a**–**e** show mean value (490 measurements for each groove) with corresponding dispersion presented with error bars, for each set of groove distance measurements. We used different colors for data in plots (**a**–**e**) for 1–4 Qt-type building blocks as well as for 3′-3′ and 5′-5′ stacking interfaces as presented in **a** d(G₂liG₄liG₂) Q4t.

will have implication not only for DNA nanotechnology in the fields of bioinspired materials and medicine, but also provide deeper understanding of fundamental properties of G-quadruplex aggregates with biological significance.

## Methods

**Oligonucleotide synthesis and purification**. All oligonucleotides (isotopically unlabeled and with residue-specific 10% $^{13}$C and $^{15}$N-labeled guanine residues as well as with incorporated c3 linkers (Spacer Phosphoramidite C3, Glen Research) and abasic residues (dSpacer CE Phosphoramidite, Glen Research)) were synthesized on K&A Laborgeraete GbR DNA/RNA Synthesizer H-8 in DMT-off mode using standard phosphoramidite chemistry. Deprotection was done at 55 °C over night with the use of aqueous ammonia which was later removed under low pressure. Samples were heated at 95 °C for 5 min in the presence of LiCl and left to cool at room temperature. Desalting was done on Amicon ultrafilter tubes at pH10, which was adjusted with the use of LiOH solution. Varian CARY-100 and CARY 3500 BIO UV–VIS spectrophotometers were used to measure absorption of desalted samples at 260 nm, which was used to calculate concentrations of stock solutions.

**Sample preparation for NMR**. NMR samples were prepared by dissolving desalted oligonucleotides in 350 µL of 90% $H_2O$/10% $D_2O$ solution, in the presence of different concentrations of KCl. KPi buffer (pH 6.8) was added to final 10 mM concentration to samples used for NMR and UV melting experiments. Oligonucleotides were folded via methods 1, 2, 3, 4, and 5 (vide infra). Sample with Q2k G-quadruplex used for $D_2O$ exchange experiment was prepared by lyophilization of previously prepared NMR sample folded via method 4 and subsequent dissolving in 100% $D_2O$.

*Method 1*. This method can be used in order to check the ability of selected oligonucleotide for G-wire formation. Annealing in the presence of relatively high concentration of KCl leads to thermodynamically favored structures, G-wires. Method 1 consists of annealing of oligonucleotide in 100 mM KCl (and 10 mM KPi buffer (pH 6.8) for NMR and UV melting experiments). The annealing procedure includes heating the samples in water bath at 90 °C for 10 min, followed by slow cooling in water back to room temperature (~8 h).

*Method 2*. This method enables observation of G-quadruplexes, which are formed at the beginning of d($G_2AG_4AG_2$) G-wire formation. Before addition of cations, sample solutions as well as KCl solution were cooled to 4 °C in order to decrease the rate of multimerization induced by addition of KCl. Different amounts of cooled KCl solution (to achieve final 100, 80, 60, 40, 20, 9, and 3 mM concentrations) were added to cooled sample solutions on which we monitored changes by NMR. Using this method we demonstrated that Q1k forms first in a wide range of KCl concentrations (3 upto 100 mM) and is always transformed into Q2k. Six days after addition of 3 mM KCl into cooled oligonucleotide solution we observed formation of next G-quadruplex in G-wire formation, i.e., Q2t. In order to increase rate of Q2t formation we increased the temperature to 25 °C and monitored changes by NMR on a new sample. Therefore, to demonstrate the Q1k→Q2k→Q2t transition, we prepared one sample at 25 °C and added KCl solution (at room temperature) to final 3 mM KCl concentration just before acquisition of NMR spectra.

*Method 3*. This method leads to sample, where Q1k is the dominant G-quadruplex structure immediately after sample preparation. At the same time, Q1k is stable (i.e., Q2k is forming slowly) over few days, which is needed to get high-quality NMR data for structural determination Samples were quenched (quenching leads to kinetically governed G-quadruplexes) in 3 mM KCl. The quenching procedure consists of heating the sample in water bath at 90 °C for 10 min followed by fast cooling on ice.

*Method 4*. Q2k is the dominant G-quadruplex structure. Since Q2k is also kinetically controlled G-quadruplex, we used same method for sample preparation as for Q1k, but in the presence of higher concentration for KCl, which we showed that favors Q2k formation. Samples were quenched (as described in "Method 3" section) in the presence of 60 mM KCl.

*Method 5*. As already established annealing of the sample leads to thermodynamically favored structures. We selected 15 mM KCl as optimal KCl concentration in order to get sample, where Q2t is the dominant G-quadruplex structure present in the sample. In lesser extend Q4t is also present in solution. Samples were annealed (as described in "Method 1" section) in between 15 and 20 mM KCl.

*DOSY sample preparation*. Since diffusion of a molecule depends also on viscosity of its solution, we prepared one sample and measured all DOSY spectra on the same sample with 1.0 mM oligonucleotide and 15 mM KCl concentrations, in 90% $H_2O$/10% $D_2O$ at 25 °C. DOSY of Q1k was measured 3 h after addition of KCl into

oligonucleotide solution at 4 °C (stored in the fridge). DOSY of Q2k was measured on the 4th day following addition of KCl. Meanwhile, sample was stored in the fridge at 4 °C. To get DOSY of Q2t, we annealed the sample using annealing procedure described in "Method 1" section.

*Phase diagram sample preparation*. In order to get phase diagram, we have prepared five samples with different oligonucleotide concentration, i.e., 0.1, 0.25, 0.5, 1.0, and 2.0 mM oligonucleotide concentration. We added KCl into oligonucleotide solutions to final 3 mM KCl concentration followed by annealing, using the same annealing procedure as described in "Method 1" section. We continued with titration of KCl to final 15, 40, 60, 80, 100, and 300 mM concentrations. At each KCl concentration we have acquired 1D $^1$H NMR spectra.

**NMR spectroscopy**. NMR experiments were recorded on Agilent Technologies DD2 600 MHz and VNMRS 800 MHz NMR spectrometers equipped with triple-resonance cryogenic probes or Bruker AVANCE NEO 600 MHz NMR spectrometer equipped with quadruple-resonance cryogenic probe at 0 and 25 °C unless stated otherwise. The double-pulsed field gradient spin echo (DPFGSE) and excitation sculpting (ES) pulse sequences were used for suppression of water signal. The translation diffusion coefficients were obtained on Bruker spectrometer with the use of stimulated echo sequence using bipolar gradients (STEbp). Guanine H1 protons were identified with the use of 1D $^{15}$N-edited heteronuclear single quantum correlation (HSQC) experiment on $^{13}$C and $^{15}$N site-specifically labeled samples. Guanine H8 protons were identified with the use of 2D $^{13}$C-edited HSQC experiment on $^{13}$C and $^{15}$N site-specifically labeled samples. 2D Nuclear Overhauser Effect SpectroscopY (NOESY) with mixing times ($\tau_m$) 150 and 250 ms were used for structural determination of Q1k, Q2k, and Q2t G-quadruplexes.

*PAD experiments*. We analyzed the effect of different KCl concentrations on G-quadruplex multimerization by monitoring changes with a set of 1D $^1$H NMR spectra acquired within the first 17 h after addition of salt. 1D $^1$H NMR spectra were acquired every 15 min (for 17 h) with the use of pre-acquisition delay (PAD) on VNMRS 800 MHz NMR spectrometer.

**UV spectroscopy**. UV melting experiments were performed on Varian CARY 3500 UV–VIS spectrophotometer with the Cary Win UV thermal program in 1.0 mm path-length cells on samples prepared via method 1 in 10 mM KPi buffer (pH 6.8). A blank sample containing only 100 mM KCl and 10 mM KPi buffer (pH 6.8) was used for baseline correction. The temperature was increased/decreased between 5 and 95 °C with the rate of 0.1 °C min$^{-1}$. Absorbance was measured at 295 nm. Mineral oil and fixed cuvette caps were used to prevent evaporation of the samples at higher temperature. To avoid condensation at lower temperature, we applied the stream of nitrogen. First derivative of $A_{295}$ versus temperature plot was used for estimation of mid-transition temperatures ($T_{1/2}$).

**Circular dichroism (CD) spectroscopy**. CD spectra were measured on Applied Photophysics Chirascan CD spectrometer at 25 °C from 200 to 320 nm. A blank sample containing only KCl (3, 15, 60, or 100 mM concentrations, depending on the sample type) was used for baseline correction. CD spectra were measured on 1.0 mM samples (prepared via method 1, 3, 4, and 5) in 0.1 mm path-length quartz cells.

**Native PAGE**. A temperature controlled vertical electrophoretic apparatus and TBE buffer were used for native PAGE gel (15%) electrophoresis. The temperature of cooler was 5 °C for both gels. G-wire samples were deposited on gel at 0.25 mM concentration per strand whereas of Q1k, Q2k, and Q2t G-quadruplexes were deposited on gel at both 0.10 and 0.25 mM concentration per strand (1 and 2.5 nmol). Prior to loading, samples were diluted with blank solution containing the same concentration of KCl as used in individual sample. PAGE were run in the presence of 0 and 25 mM KCl in the gel as well as in running buffer for G-quadruplex and G-wire samples, respectively. Prior to loading Ficoll was added to the samples. We used Thermo Scientific GeneRuler Ultra Low Range DNA Ladder (from 300 to 10 bp) as standard for size. Electrophoresis with G-quadruplex and G-wire samples were run at 150 V for 2.5 h and 3 h, respectively. DNA was visualized by Stains-All (Sigma-Aldrich) staining.

**Structure calculations**. We calculated the structures of Q4t G-quadruplexes formed by d($G_2AG_4AG_2$) oligonucleotide using the simulated annealing (SA) simulations based on NMR-derived restraints. The SA simulations were performed using CUDA version of pmemd module of AMBER 20 program suits[42] and force field with refinements as described previously[29]. Using the leap module of AMBER 20 program suits we generated the initial extended single-stranded DNA structure. We folded Q2t G-quadruplex using hydrogen bond restraints (force constant 40 kcal mol$^{-1}$ Å$^{-2}$) and NOE-derived distance restraints for 3′-3′ stacking interface (force constant 40 kcal mol$^{-1}$ Å$^{-2}$) obtained from 2D NOESY spectra ($\tau_m$ 150 and 250 ms), recorded on sample folded via method 5. Based on intensity of respective intra-residue H8-H1′ NOE cross-peaks, glycosidic torsion angles of all residues were restrained (force constant 80 kcal mol$^{-1}$ Å$^{-2}$) to *anti* region (200–280°). The

remaining torsion angles and phase angles of pseudorotation were not restraint. Q4t G-quadruplex structure was built by 5′-5′ stacking of two Q2t G-quadruplexes and added distance restraints on stacking interface. The prmtop and prmcrd files of Q4t were generated using the leap module of AMBER 20 program suits. To remove any potential clashes generated when manually building Q4t, the model underwent unrestrained energy minimization with 10,000 steps of steepest descend method. We calculated also the models for Q4t G-quadruplexes formed by d($G_2$liG$_4$liG$_2$), d($G_2$TG$_4$TG$_2$), d($G_2$CG$_4$CG$_2$), and d($G_2$abG$_4$abG$_2$) oligonucleotides using the same restraints and folding procedure as described for d($G_2$AG$_4$AG$_2$) Q4t. Force field parameters for c3 linker and abasic residues were derived from RESP ESP charge Derive (R.E.D.) Server[43]. We calculated a total of 100 structures for all five Q4t models in 500 ps long NMR-restrained SA simulations using generalized Born implicit solvent model and random starting velocities. The cut-off for non-bonded interactions was 5000 Å and the SHAKE algorithm was used for hydrogen bonds with 1 fs time steps. The SA simulation was as follows: in 0–100 ps the temperature was raised from 300 to 450 K. In the following 300 ps the temperature was scaled down to 300 K and held constant at 300 K for the next 50 ps. The temperature was reduced from 300 to 0 K in the last 50 ps. All calculated structures were subjected to restrained energy minimization with 10,000 steps of steepest descend method. Families of ten structures for all five Q4t models were selected based on the smallest restraints violation and lowest energy. The structures were visualized and prepared using Pymol software.

**Molecular dynamics (MD) simulations**. We performed 10 ns long MD simulations with 1 fs time step on all five Q4t models using NMR-derived restraints as described for SA simulations. The temperature program during MD simulation was as follows: in 0–0.5 ns the temperature was raised from 300 to 450 K. In the next 6.3 ns we scaled the temperature down from 450 to 300 K and held it constant at 300 K for next 3 ns. The temperature was decreased from 300 to 300 K in the next 0.1 ns and further decreased to 0 K in the last 0.1 ns of the MD simulation. We analyzed groove dimension between P–P atoms of neighboring loop residues of all five Q4t models. Analysis of dimension of grooves was performed for the first 9.8 ns of MD simulation on corresponding trajectory files of all five Q4t models using cpptraj module of AMBER 20 program suit. The groove dimension data was analyzed using Origin software.

**DLS measurements**. DLS measurements were performed using a frequency doubled Nd:YAG (532 nm) laser as the light source and a ALV-7002 Multiple tau digital real time correlator. Experiments were conducted at room temperature using samples with 1.0 mM oligonucleotide concentration. The experimental setup of DLS was described in detail previously[25]. The two-exponential decay function was fitted to the obtained intensity autocorrelation function g$_2$(t)[25]. The slow diffusive mode (diffusion coefficient $D_s \approx 10^{-12}$ m$^2$ s$^{-1}$) was assigned to large, unspecific clusters, which were observed in solutions of various polyelectrolytes[44–46], but are not connected to G-quadruplex formation. The fast diffusive mode (diffusion coefficient $D_f \approx 10^{-10}$ m$^2$ s$^{-1}$), on the other hand, was assigned to the diffusion of individual G-quadruplexes and quadruplex assemblies. To calculate the size of the quadruplexes from the $D_f$ values we estimated the diameter of the G-quadruplexes to be 3.0 nm, which accounts for the side loops of the quadruplex and its hydration sphere[25]. For long quadruplexes ($L \geq 2d$: longer than 6 nm and with the diffusion coefficient smaller than $1.04 \times 10^{-10}$ m$^2$ s$^{-1}$) we used the hydrodynamic model for rod-like particles, developed by Tirado and de la Torre[36]. Effective lengths and $n$ for the longest two d($G_2$abG$_4$abG$_2$) and d($G_2$AG$_4$abG$_2$) G-wires could not be accurately calculated since $L/d$ ratio is >30, which exceeds the range of theory validity. The number of stacked G-quadruplexes ($n$) was calculated based on average stacking distance of 0.34 nm and our G-quadruplex folding model. Furthermore, it should be noted that DLS signal favors larger species as the intensity of scattered light scales with the square of the scattering volume.

**Sample preparation for AFM**. Sample solution for AFM imaging was prepared by diluting 1.0 mM d($G_2$AG$_4$AG$_2$) solution with 100 mM KCl to very low oligonucleotide concentrations (2 and 0.2 μM) before depositing on mica surface. Deposition of 2 μM d($G_2$AG$_4$AG$_2$) concentration resulted in formation of G-wire networks, whereas lower 0.2 μM concentration allowed visualization of individual G-wires. To ensure better adhesion of the G-wires into the substrate, freshly cleaved V-1 muscovite mica was treated before deposition by a saturated solution of MgCl$_2$, rinsed with miliQ water and dried in ambient conditions. Next, 5 μL of the G-wire diluted solution was deposited into the pre-treated substrate and left to adsorb for 15 min. Afterwards, the excess material is removed by rinsing the substrate with 1 mL of miliQ water. The sample is then left to dry for at least one day at ambient conditions before imaging. For d($G_2$AG$_4$CG$_2$) we used same protocol as described for 0.2 μM d($G_2$AG$_4$AG$_2$). In addition, we deposited 0.2 μM of d($G_2$AG$_4$CG$_2$) on muscovite mica, which was not pre-treated with MgCl$_2$.

**AFM imaging**. Atomic force microscopy (AFM) imaging was performed in tapping mode using a Nanoscope IIIa-MultiMode AFM (Digital Instruments, Santa Barbara, CA) equipped with the E (10 μm) scanner. We used silicon cantilevers (Bruker OTESPA-R3), with nominal resonance frequency 300 kHz and nominal tip radius of 7 nm.

Height and length distribution where obtained with ImageJ and the Ridge Detection plugin (https://imagej.net/Ridge_Detection) based on the algorithm for detecting ridges and lines described by Steger[47–49]. The height and length histograms for d($G_2$AG$_4$AG$_2$) G-wires were obtained by detecting all G-wires ($N = 907$) in eight AFM images (Supplementary Fig. 1). The height and length histograms for d($G_2$AG$_4$CG$_2$) G-wires were obtained by taking into account all detected G-wires ($N = 3766$) in 12 AFM images (Supplementary Fig. 24a). In case of ($G_2$AG$_4$CG$_2$) G-wires deposited on mica, which was not pre-treated with saturated solution of MgCl$_2$, the height and length histograms were obtained by taking into account all detected G-wires ($N = 2646$) in 20 AFM images (Supplementary Fig. 24b). One should note that the actual G-wire lengths are around 10 nm shorter than what is observed by AFM due to the finite tip size effect[50].

**Sample preparation for SEM and TEM**. Sample concentration used for electron microscopy imaging was the same as for NMR and DLS experiments. G-wire samples were deposited on graphene oxide on lacey carbon 300 mesh copper TEM grids (Structure Probe, Inc., PA, USA). We deposited 10 μL of the sample (1.0 mM oligonucleotide and 100 mM KCl concentrations) on the TEM grid, waited 1 min for adsorption and removed the excess with lint free lens tissues. We washed the sample by depositing 10 μL miliQ water, waiting 1 min and removing the excess with lint free lens tissues. The washing procedure was repeated five times.

**Electron microscopy imaging**. SEM was carried out using a Zeiss Supra TM 35 VP (Carl Zeiss, Oberkochen, Germany) scanning electron microscope. The operational voltage was set to 0.7 kV with 3.5 mm working distance, 30 μm aperture size and a secondary electron detector.

TEM was performed in a Cs-corrected transmission electron microscope (CF-ARM Jeol 200). An operational voltage of 80 kV was employed. The images were taken in conventional TEM mode at substantial under- and over-focus (40 and 35 μm, respectively) to make the G-wires visible. The samples were not stained.

*Average thickness and length from SEM and TEM images*. Thicknesses and lengths of G-wires, which were nicely separated on SEM and TEM images were measured by hand using Gatan Digital Micrograph program. In total 30 measurements were done for each type of measurement and image type.

## Data availability

All relevant data supporting the findings of this study are available in the Supplementary Information. Any additional data are available from the corresponding author upon request.

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

## Acknowledgements

We gratefully acknowledge our lab technician Vesna Milek for help in the lab, especially with synthesis and purification of DNA oligonucleotides as well as Aleš Novotny, Dr. Vojč Kocman and Dr. Peter Podbevšek for their help and fruitful discussions related to structure calculations and MD simulations. We thank Department of Condensed Matter Physics at Jožef Stefan Institute for the opportunity to use their Illa-MultiMode AFM device and to Dr. Miha Škarabot for his help during AFM experiments. We thank Professor Dr. Goran Dražić from Department of Materials Chemistry at National Institute of Chemistry for productive discussion regarding sample preparation for electron microscopy and imaging. This study was supported by Slovenian Research Agency (ARRS) grants P1-0242 (J.P.), P1-0192 (I.D.O.), P2-0393 (G.K.P.) and J1-7108 (P.Š.), J1-1704 (J.P.), J7-9399 (I.D.O.) as well as by CERIC-ERIC.

## Author contributions

D.P. and P.Š. designed the experiments. D.P. synthesized and prepared DNA oligonucleotide samples, conducted NMR, CD, UV and PAGE experiments, structural calculations and MD simulations. N.S. prepared samples for AFM and carried out AFM imaging. L.S. and I.D.O. performed DLS measurements. D.P. and G.K.P. prepared samples for electron microscopy. G.K.P. conducted SEM and TEM imaging. P.Š. supervised the project. D.P., N.S., L.S., I.D.O., G.K.P., J.P., and P.Š. all contributed to analyzing and interpreting the data as well as to writing and correcting the manuscript.

## Competing interests

The authors declare no competing interests.
