## [Peer Review File · Nature Communications]

Understanding self-assembly at molecular level enables controlled design of DNA G-wires of different propertiesREVIEWER COMMENTS

Reviewer #1 (Remarks to the Author):

The authors report a significant advance in the understanding of the structure and likely folding pathway of G-wires, a supramolecular nucleic acid material. Nucleic acid nanomaterials are an area of significant research for the development of novel diagnostic and therapeutic systems. Past investigations of the structure of G-wires have used a combination of scanning probe microscopy, gel electrophoresis, dynamic light scattering and molecular dynamics simulations to make predictions as to the atom level detail of the G-DNA folds and stacking interactions. This is the first study in which NMR has successfully revealed the key inter- and intramolecular interactions within a G-wire as well as a systematic assessment of the sequence elements that influence stable self-assembly. The potential to control the morphology of the G-DNA folds in G-wires as well as the length of the supramolecular polymer would enable them to be used with greater predictability in nucleic acid nanostructures.

Overall, the manuscript has a logical organization with ample supplementary information that support the main claims of the authors. A systematic approach to G-DNA assembly was used in which coordinating cation concentration (potassium) and temperature conditions were varied and enabled the identification of two distinctly different folds derived from the same oligonucleotide sequence, dGGAGGGGAGG. The structures are broadly categorized as kinetic determined folds (Qk) and thermodynamic determined folds (Qt). The thermodynamic path yielded supramolecular assemblies and filamentous aggregates thereof which are characteristic of other known G-wire forming systems reported in the primary literature (AFM, TEM and SEM data). While this is not a novel observation, it is important to establish this as a common G-wire characteristic. Given that sample preparation for the various microscopy techniques influence the measurable features, the DLS analysis provides reproducible comparison of G-wire dimensions in solution with a useful distinct threshold for the upper limit of resolution. Both methods complement each other well in this study. I'll refrain from commenting in the molecular dynamics results presented as that is outside my area of expertise. The extensive NMR experiments for the characterization of Qk and Qt structures for dGGAGGGGAGG are convincing and strong support for an alternating mixed stacking arrangement (alternating head-to-head and tail-to-tail) using this oligo as a building block under the conditions used for self-assembly. I do not think the findings support the notion that this model extends to other G-wire forming oligonucleotides (eg. one G-block or two G-block sequences). However, it does present a clearly described strategy and experimental design for characterizing other G-wire forming systems.

The authors need to address the following:

Supplementary Fig 21. PAGE analysis of the extent of self-assembly as a function of sequence variation at positions 3 and 8 clearly reveal differences in the incremental size increases when comparing them to the model sequence. The substitution of pyrimidine residues at position 3 and or 8 show additional "rungs" in the ladder pattern as well as differences in mobility for highest mobility intermediates that are distinctly different compared to the parent structure. Furthermore, NMR melt data of G-wires assembled via Method 1 (Qt type) do not match the imino proton shifts and pattern for dGGAGGGGAGG.

Method of detection of DNA in PAGE analysis is not given. UV shadowing, stains all??? Also, mol equivalents of DNA loaded on gels is not given.

The effect of the linker substitutions (li) on the length of G-wire assemblies may not be solely due to relative hydrophobicity. The linker also has significantly greater rotational freedom compared to deoxyribose in the backbone that would likely impact the equilibrium distribution of the different G-DNA folds for this oligo. Please comment on this in the revision.

Supplementary Figure 8 line 11 has incorrect order for potassium concentrations. Should be 3, 15, 60 if it's consistent with how concentration favors formation of Q2k.

Page 14 of the main paper paragraph lines 6 on describe G-wires having melt high temps (80-95°C) where the NMR melt data shows disruption of supramolecular assembly at temps ranging from 35-50°C range. UV melts are showing the dissociation of G-DNA components of the G-wire. This distinction should be made.

Reviewer #2 (Remarks to the Author):

This manuscript entitled "Understanding self-assembly at molecular level enables 1 controlled design of DNA G-wires of different properties" studied DNA G-wire formation mechanism by use of NMR and CD spectroscopies as well as electron and atomic force microscopies. The series of well-designed study was mainly carried out with one G-rich oligonucleotide and others having similar sequences with the main one. Based on the results, a folding mechanism of G-wire was proposed, in which a formation of a mixed type G-quadruplex, its dimerization, a structural transition from the mixed to a parallel G-quadruplex (in the dimer form), and multimerization of the dimers. The results further suggested that the G-wire is homogeneous, and that the length is controllable by oligonucleotide sequence. The results shown in this manuscript are systematic and persuasive and should stimulate further studies and applications for G-wire of G-rich oligonucleotides. However, this reviewer suggests some improvements before publication of this manuscript in the Nature Communications.

- 1, The main results shown in Figure 2 are not easy to understand for the broad readers of the journal. Please reorganize to make them more understandable.
2. A series of the methods for the sample preparation was utilized. Please rationalize how each method leads to different structures.
3. Because of the multiple states in the structure, a phase diagram is required. For example, a phase diagram with different concentration of the oligonucleotide and the cation concentration could be useful for design of a desired G-wire.
4. Because of the sensitivity of G-quadruplex and G-wire to experimental conditions, it is required to study effects of salt (K⁺ vs. Na⁺ and Li⁺) and molecular crowding on the structure and the formation mechanism of the G-wire.
5. Figure 3: The formation of Q2k is reasonable, because of the exposed 3' G-quartet. The oligomerization of Q2t to form Q4t and Qnt is also possible to undergo spontaneously, because of the two G-quartet surfaces. On the other hand, it is required to discuss how the structural transition from Q2k to Q2t can undergo. In the same way, the structure of Q1k is not a typical G-quadruplex. Please discuss these points. Moreover, the rate determining step should be identified. If possible, a phase diagram depending on the reaction (incubation) time is better to be shown.
6. Thermal stability of the structures could be shown.
7. In the TEM, SEM, and AFM images, there are many branching points. Is it possible to control the number (or density) of the branching points by changing the preparation method and/or oligonucleotide sequence? This is also very important point to utilize G-wire in some applications.

Ljubljana, November 2nd, 2021**Response to Reviewers**

Manuscript number: NCOMMS-21-28343-T

Title: "Understanding self-assembly at molecular level enables controlled design of DNA G-wires of different properties"

Reviewer: 1

Comment 1: Supplementary Fig 21. PAGE analysis of the extent of self-assembly as a function of sequence variation at positions 3 and 8 clearly reveal differences in the incremental size increases when comparing them to the model sequence. The substitution of pyrimidine residues at position 3 and or 8 show additional "rungs" in the ladder pattern as well as differences in mobility for highest mobility intermediates that are distinctly different compared to the parent structure. Furthermore, NMR melt data of G-wires assembled via Method 1 (Qt type) do not match the imino proton shifts and pattern for dGGAGGGGAGG.

Our response: Firstly, to address additional 'rungs' in PAGE gels. We observed that already for the parent oligonucleotide d(G₂AG₄AG₂), additional bands appear in the gel, which we think do not reflect its behavior in solution, e.g., on the gel we observe band around 20 bp, which corresponds to Q2t and band between 35 and 50 bp corresponding to Q4t. Interestingly, we can also see band between 25 and 35 bp, which could correspond to Q3t structure, i.e. composed of 3 Qt-type G-quadruplexes, which we did not observe in solution. Clearly, these samples do not exhibit the same behavior in gel as in solution. Nonetheless, we still included PAGE analysis, because it can be nicely seen that all samples exhibit ladder-like pattern, which is characteristic for G-wires.

Regarding NMR melting data – imino proton shifts and patterns of different sequences do not match shifts and pattern of parent d(G₂AG₄AG₂) oligonucleotide, which is expected since each oligonucleotide has a slightly different nucleotide sequence. Even though the general Q2t fold is retained, there are still small differences between structures which result in different shielding

of corresponding imino protons and thus leading to different chemical shifts for each oligonucleotide.

However, to demonstrate that modified oligonucleotides follow the same folding pathway as parent d(G₂AG₄AG₂) (Fig. 3) as we demonstrated by adding KCl to final 3 mM concentration into solution of d(G₂I₄AG₂), d(G₂C₄AG₂), d(G₂AG₄I₂), d(G₂AG₄C₂), d(G₂I₄I₂) and d(G₂C₄C₂) oligonucleotides and following changes by NMR at 25°C. 1D ¹H NMR spectra recorded immediately after addition of KCl exhibit multiple sets of signals between δ 10.5 and 12.3 ppm, likely corresponding to H1 protons of guanines in G-quartets of Q1k, Q2k and Q2t, whereas only one set of signals is dominant in 1D ¹H NMR spectra one day later (Supplementary Fig. 23). We have now included additional text in the manuscript as well as a figure in Supplementary information (Supplementary Fig. 23).

Comment 2: Method of detection of DNA in PAGE analysis is not given. UV shadowing, stains all??? Also, mol equivalents of DNA loaded on gels is not given.

Our response: DNA was visualized using Stains-All (Sigma-Aldrich) staining. This sentence is now included in the revised Supplementary information under Methods section 1.5 PAGE. In the case of PAGE gel in Supplementary Figure 8 the DNA was loaded in 1 and 2.5 nmol amount, and in 2.5 nmol amount in the case of PAGE gel in Supplementary Figure 22. This information is now included in figure legends of corresponding supplementary figures.

Comment 3: The effect of the linker substitutions (li) on the length of G-wire assemblies may not be solely due to relative hydrophobicity. The linker also has significantly greater rotational freedom compared to deoxyribose in the backbone that would likely impact the equilibrium distribution of the different G-DNA folds for this oligo. Please comment on this in the revision.

Our response: We are grateful for this comment of the Reviewer regarding the effect of the linker substitutions (li) on the length of G-wire assemblies, which is now included in the revised manuscript.

Comment 4: Supplementary Figure 8 line 11 has incorrect order for potassium concentrations. Should be 3, 15, 60 if it's consistent with how concentration favors formation of Q2k.

Our response: In Supplementary Figure 8 line 11 we have wrote that: 'Prior to loading, Q1k, Q2k and Q2t samples were diluted with a blank solution containing 3, 60 and 15 mM KCl, respectively.' The order of potassium concentration is correct since we used 3 mM KCl

concentration for structural studies of Q1*k* (method 3, Methods), 60 mM KCl concentration for structural studies of Q2*k* (method 4, Methods) and 15 mM KCl concentration for structural studies of Q2*t* (method 5, Methods). We chose these conditions as optimal for structural determination since desired structure (Q1*k*, Q2*k* or Q2*t*) is the dominant species present in the sample over the time period required for the acquisition of high-resolution 2D NMR spectra, needed for structural elucidation. We have now included more detailed description of the rationale behind specific methods used for sample preparation in the revised Supplementary information under Methods section 1.1 Sample preparation for NMR.

Comment 5: Page 14 of the main paper paragraph lines 6 on describe G-wires having melt high temps (80-95°C) where the NMR melt data shows disruption of supramolecular assembly at temps ranging from 35-50°C range. UV melts are showing the dissociation of G-DNA components of the G-wire. This distinction should be made.

Our response: Indeed, UV melting curves of G-wires exhibit one transition, which corresponds to dissociation of Q2*t* as shown by NMR. Therefore, the transition observed by the UV melting experiment does not correspond to the dissociation of whole G-wires, but rather to dissociation of their smallest stable unit, namely Q2*t* G-quadruplex. We have now included a clearer description in the revised version of the manuscript.

Reviewer: 2

Comment 1: The main results shown in Figure 2 are not easy to understand for the broad readers of the journal. Please reorganize to make them more understandable.

Our response: The results shown in Figure 2 were reorganized. We hope that Figure 2 will now be easier to understand for broad readers of this journal.

Comment 2: A series of the methods for the sample preparation was utilized. Please rationalize how each method leads to different structures.

Our response: We have now included a more detailed rationalization of how different methods lead to the formation of different structures in Supplementary information under Methods 1.1. Sample preparations for NMR.

Comment 3: Because of the multiple states in the structure, a phase diagram is required. For example, a phase diagram with different concentration of the oligonucleotide and the cation concentration could be useful for design of a desired G-wire.

Our response: We are grateful Reviewer for suggesting preparation of phase diagram, which really nicely shows the distribution of structures present at certain conditions. We have now made a phase diagram with different $d(G_2AG_4AG_2)$ oligonucleotide and cation concentrations at 25°C in the equilibrium state (after annealing).

Supplementary Figure 20. Simplified phase diagram showing how oligonucleotide and K^+ concentration influence G-wire self-assembly. At conditions represented with black dots, we have acquired 1D 1H NMR spectra in order to obtain information about structures present in solution at 25°C and equilibrium state. Sample preparation is described in Methods (Phase diagram sample preparation).

The simplified phase diagram was obtained as follows:

Phase diagram sample preparation

In order to get phase diagram, we have prepared five samples with different oligonucleotide concentrations, i.e., 0.1, 0.25, 0.5, 1.0 and 2.0 mM. We added KCl into oligonucleotide solutions to final 3 mM KCl concentration followed by annealing, using the same annealing procedure as described in Method 1 (Methods). We continued with titration of KCl to final 15, 40, 60, 80, 100 and 300 mM concentrations. At each KCl concentration, we have acquired 1D ^1H NMR spectra.

All these data are now included in the revised Manuscript and Supplementary information.

Comment 4: Because of the sensitivity of G-quadruplex and G-wire to experimental conditions, it is required to study effects of salt (K^+ vs. Na^+ and Li^+) and molecular crowding on the structure and the formation mechanism of the G-wire.

Our response: G-quadruplexes and their nanostructures are indeed highly dependent on the type of present (monovalent) cations as well as molecular crowding conditions, which can cause the formation of completely different structures. To see how the type of cation affects G-wire formation, we prepared 1 mM $d(\text{G}_2\text{AG}_4\text{AG}_2)$ samples in the presence of 100 mM LiCl and 100 mM NaCl in 90% $\text{H}_2\text{O}/10\%$ D_2O via method 1 (Methods). In general, it is known that Li^+ ions do not promote G-quadruplex formation. In full agreement, 1D ^1H NMR spectrum of $d(\text{G}_2\text{AG}_4\text{AG}_2)$ in the presence of LiCl does not exhibit any signals in the H1 region of spectrum characteristic for G-quadruplex and G-wire formation (see Figure below).

1D ^1H NMR spectra of 1.0 mM $d(\text{G}_2\text{AG}_4\text{AG}_2)$ in the presence of 100 mM (a) KCl, (b) NaCl and (c) LiCl. Spectra were acquired at 600 or 800 MHz, 1.0 mM oligonucleotide, 100 mM KCl, 100 mM NaCl and 100 mM LiCl in 90% $\text{H}_2\text{O}/10\%$ D_2O . Samples were prepared via method 1 (Methods).

In contrast, 1D ^1H NMR spectrum of $d(\text{G}_2\text{AG}_4\text{AG}_2)$ in the presence of NaCl exhibits a broad hump between δ 10.0 and 11.5 ppm, which could indicate the formation of G-quadruplex based structures, potentially G-wires. It is interesting to note the presence of sharp signals present in the H8 region of 1D ^1H NMR spectrum between δ 7.5 and 8.5 ppm, which in contrast to solely broad signals observed in the same part of the spectrum of the sample in the presence of 100 mM KCl suggest the presence of smaller structures as well. However, to further test if structures formed in the presence of NaCl are indeed G-wires, we performed an NMR melt between 25 and 85°C (see Figure below). With increasing temperature, we observe a small decrease in intensity of broad hump between δ 10.0 and 11.5 ppm, which is barely detected at 85°C. In contrast to the sample with 100 mM KCl, during NMR melt, no sharp signals suggesting the presence of smaller G-quadruplexes are observed for the sample with 100 mM NaCl. This could indicate the collective transition of G-wires/G-quadruplexes, where energy for individual G-quadruplex formation is so high that once they are formed, they immediately stack further to G-wires, or alternative structures could be formed in the presence of NaCl.

NMR melting profiles of 1.0 mM $d(G_2AG_4AG_2)$ in the presence of 100 mM (a) KCl and (b) NaCl. Spectra were acquired at 600 or 800 MHz, 1.0 mM oligonucleotide, 100 mM KCl or NaCl, 10 mM KPi and NaPi buffer (pH 6.8) in 90% $H_2O/10\%$ D_2O . Samples were prepared via method 1 (Methods).

To simulate molecular crowding conditions, we prepared 1.0 mM $d(G_2AG_4AG_2)$ in the presence of 3 mM KCl and 10% wt PEG 8000 (M_w 9000 g/mol) in 90% $H_2O/10\%$ D_2O and annealed it. Comparison of H1 region of 1D 1H NMR spectrum of a sample with PEG with the sample without PEG suggests that mechanism of G-wire formation is retained (namely, signals of Q1k and Q2t can be observed). As expected, based on the known literature data, PEG shifts the equilibrium towards the formation of all-parallel G-quadruplexes, therefore to Q2t in this case.

H1 region of 1D 1H NMR spectra of 1.0 mM $d(G_2AG_4AG_2)$ in the presence of 3 mM KCl (a) without and (b) with PEG. Spectra were acquired at 800 MHz, 1.0 mM oligonucleotide, 3 mM KCl and (b) 10% wt PEG 8000 (M_w 9000 g/mol) in 90% $H_2O/10\%$ D_2O . Samples were annealed as described in method 1 (Methods).

To confirm or disapprove our hypothesis regarding structures present in the sample with NaCl, as well as to estimate how PEG influences the length distribution of formed G-wires, a whole new study should be performed, which we feel is out of the scope of this paper. Therefore if possible, we would prefer not to include it in the present paper.

Comment 5: The formation of Q2k is reasonable, because of the exposed 3' G-quartet. The oligomerization of Q2t to form Q4t and Qnt is also possible to undergo spontaneously, because of the two G-quartet surfaces. On the other hand, it is required to discuss how the structural transition from Q2k to Q2t can undergo. In the same way, the structure of Q1k is not a typical G-quadruplex. Please discuss these points. Moreover, the rate determining step should be identified. If possible, a phase diagram depending on the reaction (incubation) time is better to be shown.

Our response: We are glad about this comment since we are actually already preparing a separate paper on this topic. Structural transition from Q2k to Q2t is actually a two-step process, which occurs through long-lived G-quadruplex intermediate, Q2i. We have quite a lot of data on Q2i structure and also on oligonucleotide sequence requirements, which are needed in order for this structural transition to occur. Therefore, we would like to publish this data in a separate paper, which will be written more from G-quadruplex structural point of view and therefore it will probably be of interest for a different group of readers as compared to this paper. Furthermore, we plan to include a discussion about unusual structure of Q1k and some other data we have in a separate paper as well. However, we have now included the following sentence in the revised manuscript: 'Rearrangement from Q2k into Q2t occurs via long-lived intermediate, which will be described in detail in a separate study'.

Regarding the rate determining step of $d(G_2AG_4AG_2)$ self-assembly, we think it varies with temperature as can be seen from plots in Supplementary Fig. 14c,d. At 0°C, formation of Q2t determines the overall rate of self-assembly, since the system does not have enough energy for an immediate structural transition of Q2k to Q2t, whereas at 25°C and above, structural transition occurs rather quickly, therefore the formation of Q2k, which depends on cation concentration is the rate-determining step of $d(G_2AG_4AG_2)$ G-wire self-assembly. We have now included this information in the revised manuscript as well as in Supplementary Fig. 14 legend.

Comment 6: Thermal stability of the structures could be shown.

Our response: We have performed thermal stability studies of oligonucleotides $d(G_2AG_4AG_2)$, $d(G_2IG_4AG_2)$, $d(G_2CG_4AG_2)$, $d(G_2AG_4IG_2)$, $d(G_2AG_4CG_2)$, $d(G_2IG_4IG_2)$, $d(G_2CG_4CG_2)$, $d(G_2IG_4CG_2)$

and $d(G_2CG_4TG_2)$ by NMR as well as UV. These results are shown in Supplementary information as Supplementary Figs. 28, 29 and 31. In addition, we have done NMR melt on a sample with Q2t and Q4t present in solution (Supplementary Fig. 30), where it can be seen that the thermal stability of these structures greatly depends on the concentration of KCl (in case of 100 mM KCl (G-wire samples, Supplementary Fig. 28) we observed that Q2t is stable at 85°C. In contrast, in the case of 25 mM KCl (sample with Q2t and Q4t, Supplementary Fig. 30) Q2t is completely unfolded at the same temperature). We have now included this information that the thermal stability of these structures greatly depends on the concentration of KCl in the revised manuscript.

In order to get samples where Q1k, Q2k and Q2t are dominant structures, we need to use solution conditions with different KCl concentration, which would affect their thermal stability, so the results would not be comparable.

Comment 7: In the TEM, SEM, and AFM images, there are many branching points. Is it possible to control the number (or density) of the branching points by changing the preparation method and/or oligonucleotide sequence? This is also very important point to utilize G-wire in some applications.

Our response: Based on determined structures of our Qnt G-wires, we think that, what can be seen as branching points in TEM and SEM images are actually G-wires which are deposited randomly on each other during the drying process. Therefore TEM and SEM images (Supplementary Figs. 3, 4, 26 and 27) show some individual G-wires as well as some G-wires which seem to be branching but are likely just deposited on top of each other. On the other hand, AFM images of G-wires obtained at 0.2 μ M oligonucleotide concentration (Figs. 1c and 4b and Supplementary Figs. 2a and 25a,b) show only individual G-wires, whereas at 2 μ M oligonucleotide concentration network of G-wires can be observed (Supplementary Fig. 2b). These networks are often formed by G-wires on mica support, where G-wires appear to be interconnected but are in fact located just close to each other.

Sincerely Yours,

Dr. Primož Šket

REVIEWER COMMENTS

Reviewer #1 (Remarks to the Author):

Follow up to Author's response.

Thank you for the comments and revisions to the manuscript. Overall, the additional experiments and clarifications have improved the manuscript and support publication, I am very interested in seeing the follow up work that is alluded to in the response letter.

Regarding the response to comment 1: I see your point, never-the-less you are observing a stable intermediate between Q2, Q4 etc by PAGE. Considering the additional results provided as part of the revision of the manuscript there is a potential explanation for these bands. The Q3t, Q5t bands are much less pronounced with dGGAGGGGAGG compared to the N3 and N8 sequence variants. The model presented in the manuscript indicates G-wire growth occurs via a G2t incremental increase in size via 5'-5' stacking. The portion of the quadruplex structures with parallel strand topology (G6-G10) of Q1k, Q2k and Q2t are similar with the position of A8 in the propeller loop being the difference. Given the 3'-3' stacking mechanism and the presence of Q1k and Q2t in the same sample for certain conditions the additional intervening rungs could be Q1k at that end and a free 3'end for Q2t. The phase diagram that was added indicates that Q1k and Q2t are present at the same time over a broad range of conditions. It's possible they could form a stable hybrid and these intermediates could be due to the greater stability of the edge loop of Q1k with Py at position 3. I take from the response to the 2nd reviewer aspects of the folding transition from are currently being investigated. That would be an interesting aspect of G-wires self-assembly process. I am curious to see how the kinetics of the Q1k,Q2k to Qt for the edge loop and propeller loop variants assembled via method 2 to compare with that of G2AG4AG2.

It would be very helpful to have all figure captions include the assembly method used.

Reviewer #2 (Remarks to the Author):

This manuscript has been improved according to the reviewer's comments. Now the results lead to the conclusion and show polymorphic nature of G-quadruplex and G-wire.

One more additional comment is that the reader will be wondering how the sequence was designed. For example, why the guanine stashes are two, four, and two, why the loop length is one nucleotide, and why the loop is with adenine. Although the last one was answered by the results. Is it possible to explain why the author designed and selected this mother sequence?

We have considered all the comments and suggestions of both reviewers and made the following changes (presented in pink, previous changes are still presented in red) in the revised Manuscript.

Reviewer: 1

Comment 1: Regarding the response to comment 1: I see your point, never-the-less you are observing a stable intermediate between Q2, Q4 etc by PAGE. Considering the additional results provided as part of the revision of the manuscript there is a potential explanation for these bands. The Q3t, Q5t bands are much less pronounced with dGGAGGGGAGG compared to the N3 and N8 sequence variants. The model presented in the manuscript indicates G-wire growth occurs via a G2t incremental increase in size via 5'-5' stacking. The portion of the quadruplex structures with parallel strand topology (G6-G10) of Q1k, Q2k and Q2t are similar with the position of A8 in the propeller loop being the difference. Given the 3'-3' stacking mechanism and the presence of Q1k and Q2t in the same sample for certain conditions the additional intervening rungs could be Q1k at that end and a free 3'end for Q2t. The phase diagram that was added indicates that Q1k and Q2t are present at the same time over a broad range of conditions. It's possible they could form a stable hybrid and these intermediates could be due to the greater stability of the edge loop of Q1k with Py at position 3. I take from the response to the 2nd reviewer aspects of the folding transition from are currently being investigated. That would be an interesting aspect of G-wires self-assembly process. I am curious to see how the kinetics of the Q1k,Q2k to Qt for the edge loop and propeller loop variants assembled via method 2 to compare with that of G2AG4AG2.

Our response: We are grateful for possible explanation of additional bands observed in PAGE gel. We have now included explanation in the figure legend of Supplementary Figure 22. However, we believe that formation of such Q1k-Qnt hybrid structures is specific for gel and does not occur in solution, as demonstrated by 2D ¹³C-filtered HSQC experiments acquired on d(G₂AG₄AG₂) sample with partially isotopically labeled G1 residue, where we observe only cross-peaks corresponding to Qt-type of structures (Supplementary Figures 19b-e).

Comment 2: It would be very helpful to have all figure captions include the assembly method used.

Our response: Assembly methods are now included in the figure legends in main text as well as in supplementary information.

Reviewer: 2

Comment 1: One more additional comment is that the reader will be wondering how the sequence was designed. For example, why the guanine stretches are two, four, and two, why the loop length is one nucleotide, and why the loop is with adenine. Although the last one was answered by the results. Is it possible to explain why the author designed and selected this mother sequence?

Our response: Previously, we showed that GC ends attached to either 5'- or both 5'- and 3'-ends of oligonucleotide $d(G_2AG_4AG_2)$ hindered multimerization. In general, such type of oligonucleotides, i.e., $G_2X(G_4X)_nG_2$ (where $X = T, A, C$ and $n = 0, 1, 2, \text{ and } 4$) with 5'- and/or 3'-GC ends, were subject of many studies for their multimerization ability before. We have now included explanation on selection of parent (mother) oligonucleotide together with relevant references in the introduction part of the main text.